# Effect of Temperature, Light, and Storage Time on the Seed Germination of *Pinus bungeana* Zucc. ex Endl.: The Role of Seed-Covering Layers and Abscisic Acid Changes

**Congcong Guo** [1,2], **Yongbao Shen** [2,3,4,*] **and Fenghou Shi** [2,3,4]

1   College of Landscape Architecture, Nanjing Forestry University, Nanjing 210037,
    China; guocongcong0110@163.com
2   Collaborative Innovation Center of Sustainable Forestry in Southern China, Nanjing Forestry University,
    Nanjing 210037, China; fhshi406@163.com
3   Southern Tree Seed Inspection Center, National Forestry and Grassland Administration,
    Nanjing 210037, China
4   College of Forestry, Nanjing Forestry University, Nanjing 210037, China
*   Correspondence: ybshen@njfu.edu.cn; Tel.: +86-25-8542-7403; Fax: +86-25-8542-7402

**Abstract:** *Pinus bungeana* Zucc. ex Endl. is an endemic conifer tree species in China with high ornamental value. In order to investigate favorable conditions for seed germination and explore the germination inhibition mechanism of this species at high temperatures, the effects of temperature, light, and storage on the mean germination time (MGT), speed of germination (SG), and total germination percentage (TGP) are evaluated here. Seeds that have either been kept still or entered into a state of dormancy at high temperature are assessed here by a recovery experiment. Furthermore, the contribution of covering layers on thermo-inhibition is analyzed here, including the way they work. This has been realized by the structural observation and via the determination of the abscisic acid (ABA) content. The results show that seeds germinate to a high percentage (approximately 90%) at temperatures of 15 or 20 °C, with or without light, whereas higher temperatures of 25 or 30 °C impeded radicle protrusion and resulted in the germination percentage decreasing sharply (within 5%). Inhibition at high temperatures was thoroughly reversed (bringing about approximately 80% germination) by placing the ungerminated seeds in favorable temperatures and incubating them for an additional 30 days. Dry cold storage did little to reduce the temperature request for germination. Embryo coverings, especially the nucellar membrane, and ABA levels both had a dominant role in seed germination regulation in response to temperature. Under favorable temperature conditions, the levels of ABA significantly decreased. Germination occurred when the levels dropped to a threshold of 15 ng/g (FW (Fresh Weight)). Incubation at a high temperature (25 °C) greatly increased ABA levels and caused the inhibition of radicle protrusion.

**Keywords:** *Pinus bungeana*; germination; temperature; light; storage; covering layers; ABA

## 1. Introduction

Seed germination is where the youngest form of a plant (an embryo) resumes its normal physiological activities, and it is the starting point of growth for plants. The ability of a viable seed to germinate and exactly when it germinates is determined by a series of factors, including the inherent causes of germination and the external environmental conditions. Temperature and light are ecological factors of importance in the regulation of seed germination [1].

Temperature is one of the primary factors affecting the percentage and speed of germination, which directly works via seed imbibition and the biochemical reactions that regulate the metabolism involved in the germination process [2]. Further, most species require an appropriate temperature range or alternate temperature mode to achieve maximum germination. In weeds, the germination behavior of seeds is also related to the time of seed produced and the moment elapsed from the seed settling. This behavior is owed to the environmental conditions undergone by the mother plant during seed maturation and those undergone by the seeds after settling [3]. The germination percentage usually increases linearly with temperature up to an optimal temperature, after which the germination percentage decreases sharply [4–6]. Increased temperatures not only affect seedling growth after seed germination, but also directly affect the germination process. To prevent seedlings from being damaged after germination, physiological reactions may occur in seeds to cope with the high-temperature environment in which they are placed. This ecological requirement can be considered as an adaptation strategy to guarantee favorable conditions for seedling development and survival in some species [7]. Then, these seeds will cease germination, yet they will instantly germinate upon being exposed to suitable temperatures, a process which is called thermo-inhibition [8]. For most perennial or winter annual plants, their favorable germination temperature range is 10–20 °C, and these ecological habits are quite important for them to adapt to warm climates [9]. Sometimes, high or low temperatures also result in secondary dormancy, known as thermos-dormancy [10]. Under these circumstances, germination will not occur at any temperature, including the optimum temperature. This phenomenon is especially prevalent in annual desert plants and some Mediterranean species [11].

Similar to temperature, light is an important environmental factor that acts directly on germination, and the sensitivity of seeds to light is highly variable according to the species, where there are seeds that germinate more readily either only under light or darkness, while others are not affected by this factor [12]. Although light is not a necessary factor for germination for all species, it helps to alleviate the adverse effects of germination when the incubating temperature is higher than what is favorable [13]. Moreover, light requirements for seed germination may vary with changes in temperature. The fluctuation of temperature improves the germination of *Leptochloa chinensis* (Linn.) Nees seeds in the absence of light. The light requirements of *Vellozia* spp. seeds also can be altered by constant temperature shift regimes [14].

It has been noted that high temperatures inhibit seed germination in many species, and this thermal inhibition is usually attributed to high levels of endogenous abscisic acid (ABA), which is realized by the transcriptional activation of ABA signaling genes [15,16]. High temperatures up-regulate ABA biosynthesis genes and down-regulate catabolism genes [15]. While at supraoptimal temperatures, ABA is most likely to have a role in inhibiting germination by impeding radicle elongation, which is realized in a dose-dependent manner and may be ascribed to the hindrance of cell wall slackening through the inhibition of the synthesis and activities of cell wall degradation enzymes [17,18]. When seeds are exposed to favorable temperature conditions, or with the application of the ABA biosynthesis inhibitors (such as fluridone), changes in this phytohormone favor seed germination [19]. Additionally, under high-temperature conditions, the sensitivity of seeds to ABA increases, which is partly due to the embryo being unable to inactivate ABA and the reduced dissolved oxygen concentrations in the imbibed seeds [20]. Hypoxia interferes with the oxidative catabolism of this hormone by regulating ABA 8'-hydroxylase activity, whereas the temperature and oxygen content regulate enzyme function at the expression level of the gene that encodes it [21].

*Pinus bungeana* Zucc. Ex Endl. is a native plant in China with great ornamental value and adaptability to drought and cold climates. Therefore, it is widely used in landscaping and afforestation in the north of China. *Pinus bungeana* is propagated from seeds, yet previous studies generally have held that the seeds were dormant, and earlier research has mostly focused on seed dormancy. Dong et al. [22] indicated that the embryo appeared dormant, dominated by the megagametophyte. However, subsequently, this view was denied in the research of other investigators, where they considered that the embryo itself was non-dormant while the seed coat dominated seed dormancy,

and the permeability barrier might be one of the reasons that for the dormancy of the seeds [23,24]. Moreover, other studies have reported that the inhibitors in the seed tissues surrounded the embryo contribute most to seed dormancy. However, these researchers differ in their claims for the number of inhibitors in each tissue [25,26]. However, recent research has shown that seeds of this species are non-dormant at all, and previous findings of poor germination at 25 °C may be more accurately explained, as this temperature is not a suitable incubation condition for germination [27].

As the germination conditions of *Pinus bungeana* have not been comprehensively represented to date, knowledge of the effects of temperature on seed germination, the light requirements, and the respective interactions during germination in this pine is critical, both for practical nursery applications and for conservation. Moreover, the mechanism of germination inhibition maintenance induced by high temperatures in *Pinus bungeana* seeds has not yet been investigated, particularly in the context of the function of ABA. Given the above information, the objective of this study is to assess the effects of temperature, light, and storage on germination. In addition, we ask whether the inhibition of temperature on germination affects thermo-inhibition or thermo-dormancy? We also investigate the covering layers that inhibit seed germination at high temperatures and how (i.e., via morphology or chemicals) they inhibit germination. Finally, we examine the role ABA plays in the mechanism of thermo-inhibition.

## 2. Materials and Methods

### 2.1. Seed Materials

Pinecones were collected during October 2014, 2015, 2016, and 2017 from a natural *Pinus bungeana* forest on Xiaolong Mountain, Gansu Province, northeast China (~34.00°–34.40° N, ~105.30°–106.30° E). The seeds were separated carefully from the cones and subsequently delivered to Nanjing Forestry University. Then, damaged seeds were removed, and empties were rejected via the floating method [28]. As determined by the 2,3,5-triphenyl tetrazolium chloride (TTC) method [29], the viability of the fresh seeds collected over the different years was >90%. The seed batches of 2014, 2015, and 2016 were stored at 5 °C until the date of the established germination experiment, and the 2017 seed batch was immediately germinated.

### 2.2. The Effect of Temperature and Light on Seed Germination

Germination experiments were carried out under five different temperature conditions (10, 15, 20, 25, and 30 °C) for 30 days in either a light or dark environment. Before the germination test, the fresh *Pinus bungeana* seeds were soaked in purified water for 96 h. After that, they were put into a plastic germination box lined with moist absorbent cotton, and three samples of 100 seeds were subjected to each level of treatment. Then, the boxes were distributed in a GTOP-300D incubator (Zhejiang Top Cloud-agri Technology Co., Ltd., China) at random with a constant temperature. The light treatments maintained a photoperiod of 12 h light/12 h dark, and the dark treatment condition was achieved by covering the germination boxes with tinfoil. Under dark conditions, the germination was assessed in a room equipped with a green light. The germinated seeds were counted every alternate day until 30 days had elapsed. When the radicle emerged (≥2 mm), a seed was referred to as having germinated.

The total germination percentage (TGP) was formulated according to the equation below:

$$\text{TGP} = \frac{n}{N} \times 100\% \tag{1}$$

where $n$ is the germinated seeds in the trial and $N$ is the number of tested seeds (100).

The mean germination time (MGT) and speed of germination (SG) were determined as follows:

$$\text{MGT} = \sum \frac{n_i \times t}{n} \tag{2}$$

where $n_i$ is the number of germinated seeds at day $t$ and $n$ is the sum of germinated seeds in the trial.

$$SG = \sum \frac{n}{t} \tag{3}$$

where $n$ is the number of newly germinated seeds at day $t$.

### 2.3. The Effect of Dry Cold Storage on Germination

In October 2017, the seed batches of 2014, 2015, and 2016, which were stored for three years, two years, and one year, respectively (hereinafter called "CS-3", "CS-2", and "CS-1") were taken out from storage and used for germination tests at 10, 15, 20, 25, and 30 °C with a 12-h photoperiod. The 2017 seed batch was used as a control ("CS-0", non-cold exposure). The cultural conditions were the same as Section 2.2. Three samples of 100 seeds were subjected to each level of treatment. Parameters TGP, MGT, and SG were evaluated at the end of the incubation period.

### 2.4. Germination Recovery Experiments

To investigate whether high temperatures (25 and 30 °C treatments) induced the occurrence of secondary dormancy and the loss of seed viability, after 30 days, those ungerminated seeds of CS-3, CS-2, CS-1, and CS-0 were diverted to an optimum temperature of 20 °C. Then, germination was observed for another 30 days. After the recovery experiment, the TGP was calculated and all remaining non-germinated seeds were tested for their viability with the positive tetrazolium solution [30].

### 2.5. Embryo Coverings Contributed to High-Temperature Inhibition

To explore what led to the failure of seed germination at high temperatures, a batch of fresh seeds were exposed to an embryo covering removal test. The different treatments are annotated in Table 1. After four days of imbibition, the seeds were dissected under laminar flow and the seed parts were put into a plastic germination box lined with moist absorbent cotton in a 25 °C environment with a 12-h photoperiod. Three samples of 50 seeds were subjected to each treatment. The germinated seeds were scored daily for 30 days, and the TGP and rot rate (percentage of rotten seeds) was determined at the end of each experiment.

**Table 1.** Seeds germination after the sequential removal of tissue surrounding the embryo.

| Treatment | Germination (%) | Rot Rate (%) |
|---|---|---|
| Intact seeds | 5.33 ± 1.15 c | 10.67 ± 3.05 a |
| Seed coat cracked at the micropylar end | 11.33 ± 3.45 c | 14.00 ± 4.00 a |
| Seed coat removed (nucellar membrane intact) | 34.67 ± 5.03 b | 12.00 ± 2.00 a |
| Seed coat and nucellar membrane removed | 84.00 ± 2.00 a | 10.67 ± 3.05 a |
| Seed coat, nucellar membrane, and nucellar cap removed | 89.33 ± 3.05 a | 9.33 ± 2.00 a |
| Isolated embryos | 86.67 ± 1.52 a | 13.33 ± 1.52 a |

The same lowercase letter in the row indicates no significant difference at a 0.05 probability level.

### 2.6. Changes in Structure and ABA Content of Nucellar Membrane During Germination at Different Temperature

Three samples of 800 seeds were separately cultured at 20 °C and 25 °C, where the rest of the conditions were the same as in Section 2.5. One hundred seeds were collected every two days (seeds for scanning electron microscopy were collected every four days) after the start of incubation until 14 days had elapsed. After collection, the seeds were dissected using forceps and a tweezer. Then, the nucellar membranes of 5 seeds were directly used for scanning electron microscopy (SEM). The remaining

nucellar membranes were gathered and immediately stored at −80 °C until being used for the ABA determination analyses.

Half of the nucellar membranes (with an endosperm or seed coat for easier fixing and observation) were fixed on the sample table with a double-sided adhesive after an E-100 sputter coater (Hitachi, Japan) was used to gold-plate the surfaces of the samples. Subsequently, the surfaces were observed and pictured at 15 kV with the Quanta-200 scanning electron microscope (FEI, Eindhoven, Netherlands).

About 0.2 g of the nucellar membrane samples was homogenized in liquid nitrogen and extracted in 6 mL of 80% methanol with butylated hydroxytoluene (40 mg L$^{-1}$) for 48 h at 4 °C, and the extracts were then centrifuged for 15 min (4000 rpm, 4 °C). The supernatant was obtained via C18 Sep-Pak cartridges (Waters Corp., Milford, MA, USA), and the phytohormone fraction was eluted with 10 mL of 100% (*v/v*) methanol and then 10 mL of ether. The eluate was dried in N$_2$ and then the dried extracts were dissolved in 2 mL of phosphate-buffered saline (PBS) with 0.1% (*v/v*) Tween-20 and 0.1% (*w/v*) gelatin (pH 7.5) to determine the ABA levels by enzyme linked immunosorbent assay (ELISA) [31].

### 2.7. Statistical Analyses

Values were represented as the mean ± SD (standard deviation) for three replicates. Figures were processed by Excel (Office 2013 Pro Plus, Microsoft Corporation, Redmond, WA, USA). A one-way or two-way analysis of variance (ANOVA) was conducted by the SPSS 19.0 software package (IBM, Armonk, NY, USA), followed by Duncan's multiple range test. Here, *p*-values less than 0.05 were considered significant.

## 3. Results

### 3.1. Temperature and Light Effect on Seed Germination

A two-way ANOVA indicated that the TGP, SG, and MGT of *Pinus bungeana* seeds were all significantly different among the incubation temperatures. No significant differences in these parameters were observed between light and dark germination conditions ($p > 0.05$), and the interaction of the light condition and temperature regime was also not significant (Table 2), implying that the germination response pattern of the tested pine seeds to temperature is the same under different light conditions.

**Table 2.** Effects of temperature and light on the total germination percentage (TGP), mean germination time (MGT), and speed of germination (SG) of *Pinus bungeana* seeds.

| Source | Sum of Squares | df | Mean Square | *F*-Value | *p*-Value |
|---|---|---|---|---|---|
| TGP | | | | | |
| Light (L) | 8.533 | 1 | 8.533 | 1.446 | 0.243 |
| Temperature (T) | 54,750.867 | 4 | 13,687.717 | 2319.952 | 0.000 * |
| L × T | 9.800 | 4 | 2.450 | 0.415 | 0.796 |
| MGT | | | | | |
| Light (L) | 0.642 | 1 | 0.642 | 0.478 | 0.499 |
| Temperature (T) | 415.517 | 3 | 138.506 | 103.157 | 0.000 * |
| L × T | 2.443 | 3 | 0.814 | 0.607 | 0.620 |
| SG | | | | | |
| Light (L) | 0.009 | 1 | 0.009 | 0.342 | 0.567 |
| Temperature (T) | 173.010 | 3 | 57.670 | 2272.303 | 0.000 * |
| L × T | 0.175 | 3 | 0.058 | 2.304 | 0.116 |

* Significant at $p < 0.05$, others are non-significant.

Considering the data from the light and dark conditions, the highest germination (>90%) was achieved when the seeds were cultured for 30 days at 20 °C, and a slightly lower temperature of 15 °C was also an effective temperature for a high germination percentage (about 85%). Higher temperature regimes of 25 °C and 30 °C hindered radicle protrusion and significantly lowered the germination percentage (≤5%). No seed germinated at a temperature of 10 °C (Table 3). At 20 °C, seeds significantly showed ($p < 0.05$) the highest speed of germination (SG) (5.95% day$^{-1}$ and 5.62% day$^{-1}$, respectively) and the shortest mean germination time (MGT) (13.85 days and 13.45 days, respectively) as compared to that of seeds cultured at 15 °C. The MGTs of the 25 °C and 30 °C seeds were between 22 and 24 days, and their SGs were all <0.5% day$^{-1}$, which is significantly smaller than that at 15 °C.

**Table 3.** TGP, MGT, and SG of *Pinus bungeana* seeds under different light and temperature regimes.

| Temperature (°C) | TGP (%) | | MGT (day) | | SG (% day$^{-1}$) | |
|---|---|---|---|---|---|---|
| | Dark | Light | Dark | Light | Dark | Light |
| 10 | 0.00 ± 0.00 d | 0.00 ± 0.00 c | / | / | / | / |
| 15 | 84.00 ± 2.73 b | 87.33 ± 3.79 b | 17.20 ± 0.15 b | 17.61 ± 0.84 b | 5.11 ± 0.21 b | 5.23 ± 0.08 b |
| 20 | 91.00 ± 3.21 a | 92.67 ± 3.21 a | 13.85 ± 0.92 c | 13.45 ± 0.39 c | 5.95 ± 0.31 a | 5.62 ± 0.07 a |
| 25 | 4.33 ± 2.08 c | 5.00 ± 4.00 c | 23.39 ± 2.11 a | 22.07 ± 1.89 a | 0.18 ± 0.08 c | 0.24 ± 0.21 c |
| 30 | 0.67 ± 0.58 d | 1.00 ± 1.15 c | 24.00 ± 0.00 a | 24.00 ± 1.00 a | 0.04 ± 0.00 c | 0.04 ± 0.00 c |

The same lowercase letter in the row indicates no significant difference at the 0.05 level.

### 3.2. Effect of Dry Cold Storage on Germination

The germination of *Pinus bungeana* seeds at each temperature showed a significant difference among storage treatments, except for the 10 °C and 30 °C conditions. Seeds germinated well at 15 °C and 20 °C after the storage treatment (Figure 1). Both fresh seeds (CS-0) and seeds stored for one year (CS-1) achieved high germination rates (86.67% and 88.33% for CS-0 and 81% and 82% for CS-1 at the two temperatures, respectively), and the difference in germination between the two seed batches was not significant at the same temperature. Then, seed germination was significantly decreased as the storage time increased. After two or three years of storage (CS-2 or CS-3), the seeds exhibited a 52.67%–70.67% germination rate. However, at 25 °C, the fresh seeds (CS-0) only 9.67% germinated, which is significantly lower than that of the rest of seed batches (20% approximately). However, extremely low germination (< 5%) was induced or none of the seeds germinated when those stored seeds were incubated at the temperatures of 30 °C and 10 °C.

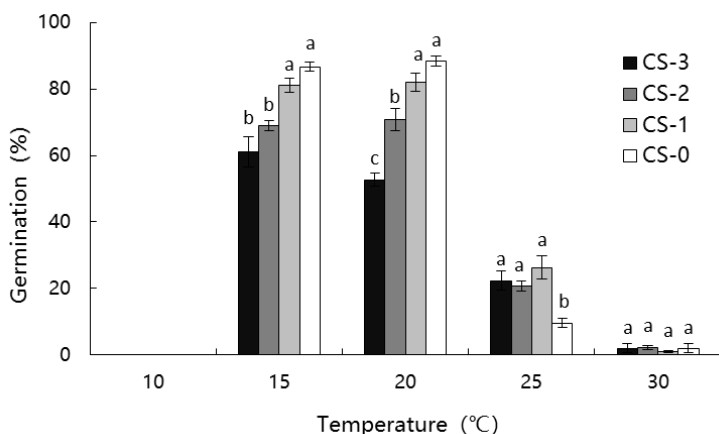

**Figure 1.** Germination of *Pinus bungeana* seeds cultured at 10, 15, 20, 25, and 30 °C. Data are the mean ± SD, and different lowercase letters within the same temperature treatment indicate significant differences at a 0.05 probability level.

Under conditions of 15 °C and 20 °C, the storage treatment had a significant effect on the MGT and SG. When incubated at 15 °C, the MGTs of fresh seeds (CS-0) and seeds stored for one or two years (CS-1 and CS-2) were all around 18 days, and were markedly smaller than that of the CS-3 seeds (19.90 days) (Figure 2a). The CS-0 and CS-1 seeds obtained a faster SG (4.95% day$^{-1}$ and 4.65% day$^{-1}$, respectively) as compared to the other two seed batches (< 4% day$^{-1}$). At the temperature of 20 °C, the SG of the CS-0 seed batch was fastest (5.51% day$^{-1}$), followed by the CS-1 and CS-2 seed batches (4.78% day$^{-1}$ and 4.43% day$^{-1}$, respectively) (Figure 2b). The seed batches of CS-0 and CS-1 had smaller MGTs (12.65 days and 13.27 days, respectively) than the remaining treatments (> 15 days) (Figure 2a).

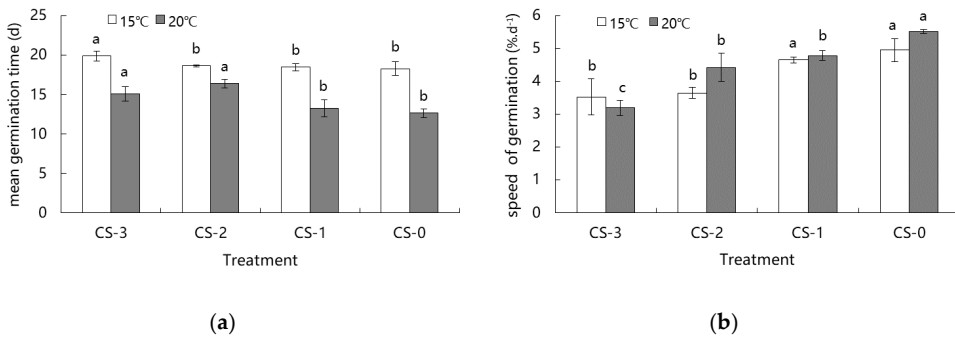

| (**a**) | (**b**) |

**Figure 2.** Mean germination time (**a**) and speed of germination (**b**) of *Pinus bungeana* seeds in response to different temperatures. Data are means ± SD and different lowercase letters in the same temperature treatment indicate significant difference at a 0.05 probability level.

### 3.3. Seed Recovery Responses

The seeds batches of *Pinus bungeana* that were stored for a different time showed lower overall germination (initial germination) when cultured for 30 days at 25 °C and 30 °C. Ungerminated seeds from these two high-temperature treatments showed a high recovery of germination (recovery percentage) when transferred to an optimum germination temperature (20 °C). The final germination rates (sum of initial germination and the recovery percentage) were all approximately 80% (except for the CS-3 treatment), which did not differ significantly among seeds stored for a different times that were transferred from 25 °C and 30 °C to the temperature of 20 °C (Figure 3). All remaining ungerminated seeds of *Pinus bungeana*, were tested with TTC and found dead after the recovery experiments (i.e., non-viable).

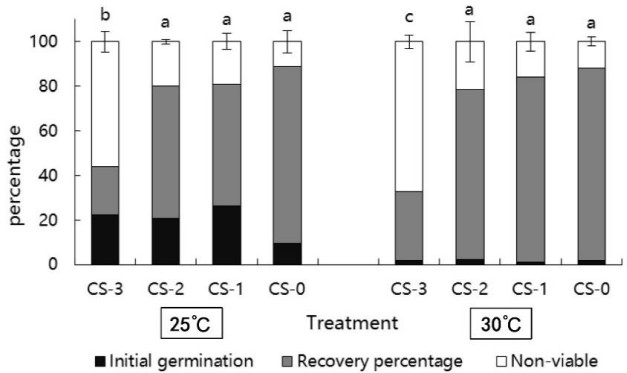

**Figure 3.** Recovery germination of *Pinus bungeana* seeds after transfer from high temperature to an optimum germination temperature. The temperature in the rectangle refers to the initial high temperature that the seeds were incubated at. The final germination (sum of initial germination and recovery percentage) is shown as the mean ± SD and different lowercase letters within the same incubation temperature indicate a significant difference at a 0.05 probability level.

### 3.4. Embryo Coverings Contributed to High-Temperature Inhibition

Through analysis of the germination after the sequential excision of the embryo coverings (Table 1), it was found that when the micropylar end of seed coats was cracked, the germination was not significantly different to that of intact ones (5.33% vs. 11.33%), while removing the whole seed coat promoted germination by 34.67%. The removal of both of the seed coats and nucellar membrane promoted even greater germination (>80%), which was not significantly different to that of the isolated embryos (86.67%).

### 3.5. Changes in Morphology of Nucellar Membrane During Germination at Different Temperature

The nucellar membrane of an imbibed *Pinus bungeana* seed comprises more than one layer of longitudinally elongated parenchyma cells (Figure 4a(C1)). At 25 °C, there were no significant changes in the morphologies of the nucellar membranes (Figure 4a(A1–A4)), which remained at their initial state at the end of the incubation. However, at 20 °C, a longitudinal tear occurred in the nuclear membranes after eight days (Figure 4b(A3,A4)). However, there were no obvious cellular differences between the nucellar membranes (their elongated cells were still staggered and closely arranged), observed by the light microscopy between seeds germinated at 20 °C (whether before or after the rupture of the nucellar membrane) (Figure 4b(B1–B4,C1–C4)) and seeds incubated at 25 °C (Figure 4a(B1–B4,C1–C4)).

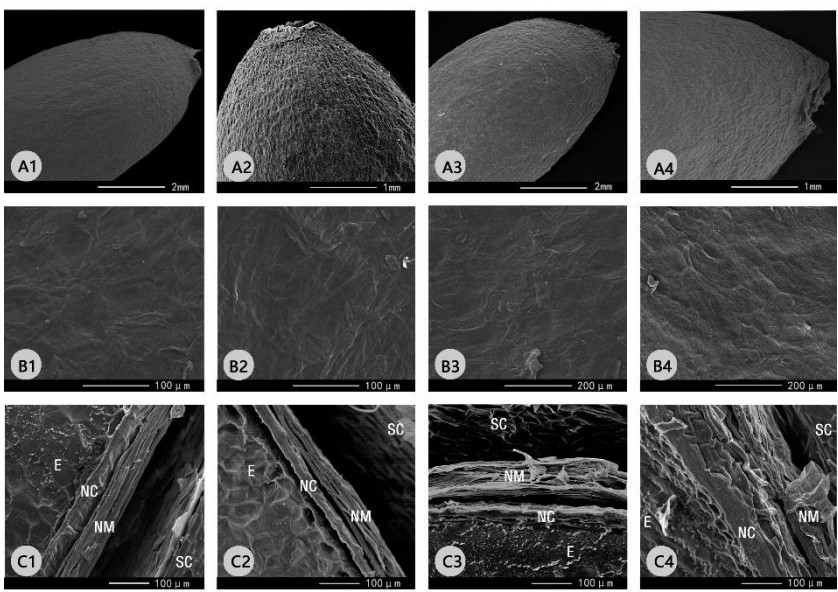

(a)

**Figure 4.** *Cont.*

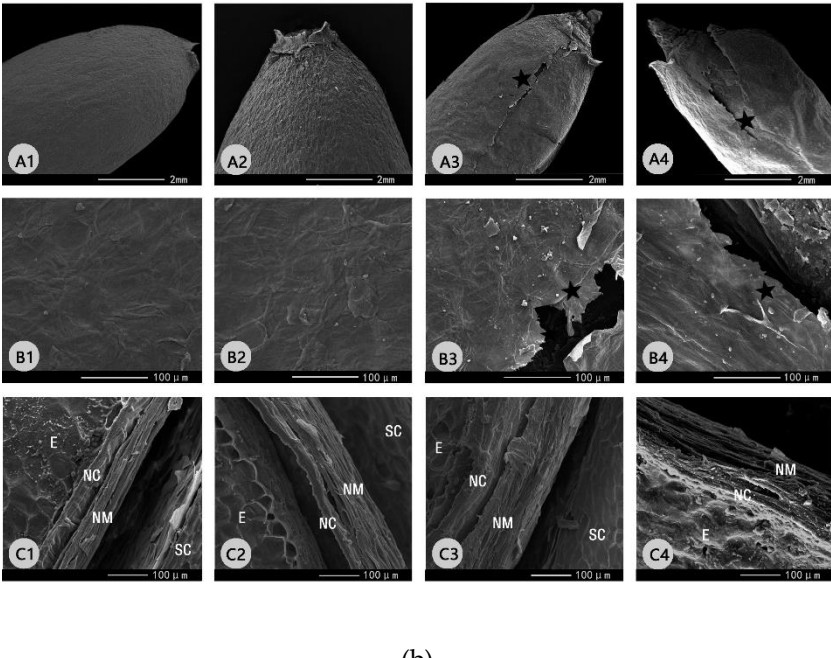

(b)

**Figure 4.** SEM micrographs of nucellar membranes during germination at 25 °C (**a**) and 20 °C (**b**). A1–A4 show most of the nucellar membranes (0, 4, 8 and 12 days). B1–B4 show the surfaces of the nucellar membranes (0, 4, 8 and 12 days). C1–C4 show the longitudinal section of the nucellar membranes (0, 4, 8 and 12 days). The rupture of nucellar membrane is marked with a black pentagram. SC, seed coat. NM, nucellar membrane. NC, nucellar cap. E, endosperm (megagametophyte).

*3.6. Changes in ABA Content of Nucellar Membrane during Germination at Different Temperature*

The nucellar membrane of the imbibed *Pinus bungeana* seeds contained 36.4 ng/g (FW) of ABA. The ABA content in the nucellar membrane at 20 °C decreased soon after the start of incubation. The values were 30.14 and 14.60 ng/g (FW) on the second and sixth days of incubation, respectively. When the seeds were incubated at 25 °C, the ABA content of the nucellar membrane decreased more slowly than that at 20 °C, and it decreased to 28.74 ng/g (FW) on the sixth day and 27.77 ng/g (FW) on the eighth day of incubation. However, after ten days of incubation, the ABA content increased to 31.93 ng/g (FW), which was comparable to the initial level. Further, 14 days after incubation, the amount of ABA in the nucellar membrane of seeds at 25 °C was twice as high as that measured in the nucellar membrane from those seeds incubated at 20 °C (Figure 5).

Figure 6 shows the relationship between the germination and ABA levels of the nucellar membrane in the seeds. At 20 °C, when the ABA content reduced to about 15 ng/g (FW), the seeds began to germinate, and germination gradually increased after that. While at the higher temperature of 25 °C, those non-germinated seeds contained more than 25 ng/g (FW) of ABA.

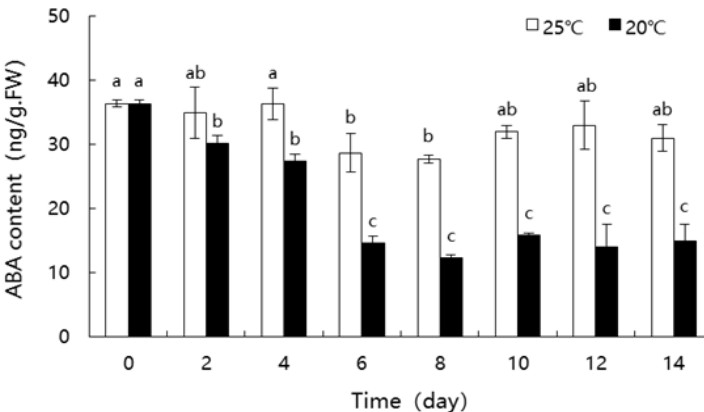

**Figure 5.** Changes in the abscisic acid (ABA) contents of the nucellar membrane in *Pinus bungeana* seeds at 20 °C and 25 °C. Data are represented as the mean ± SD, and the different lowercase letters in the same temperature treatment indicate a significant difference at a 0.05 probability level.

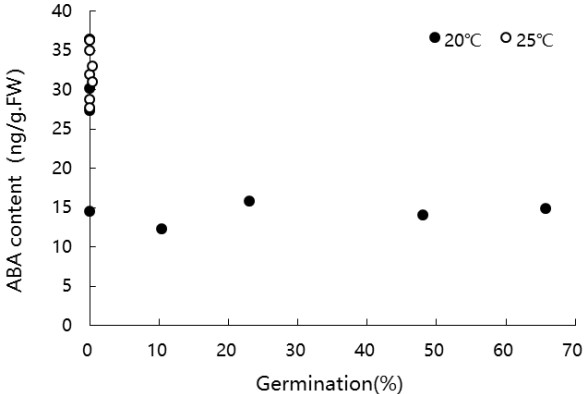

**Figure 6.** Relationship between germination percentages and ABA contents of the nucellar membrane in *Pinus bungeana* seeds. Data used were those at 0, 2, 4, 6, 8, 10, 12, and 14 days for seeds cultured at 20 °C and 25 °C, respectively.

## 4. Discussion

### 4.1. Seed Germination Responses

Light is not an environmental factor that is necessary for germination, but it is generally considered as a promotor of seed germination. Escudero et al. [32] noted that light was notably favorable to *Pinus uncinata* Ramond ex DC. seed germination. The germination of *Pinus brutia* Ten. and *Pinus halepensis* Mill. seeds have also always promoted by white light [33]. Different responses to light have been observed for *Pinus bungeana* seeds. In our study, germination tests in light/dark and dark regimes alone showed that the germination of *Pinus bungeana* seeds is equally feasible in dark conditions, and the total percentage of germination, germination speed, and mean time they took to complete germination remained unchanged, meaning that there were no significant differences between seeds cultured in light or those exposed to darkness alone, indicating that the germination of *Pinus bungeana* seeds has no rigid demand for light conditions. Similarly, the *Pinus nigra* J. F. Arnold [34] seeds germinate smoothly in both light and darkness. Therefore, the sensibility of pine seeds to light is determined by the species.

In contrast, temperature markedly affected the germination of *Pinus bungeana* seeds. In general, most subtropical and temperate seeds are sensitive to temperature variations between 20 °C and 30 °C for their germination. Both *Ginkgo biloba* L. [35] and *Wollemia nobilis* W. G. Jones, K. D. Hill, and J. M. Allen [36] seed germination proceeds most rapidly at the temperature range of 25–30 °C.

For the *Pinus lambertiana* Douglas and *Pinus koraiensis* Siebold and Zucc. [30] seeds, the germination test should be conducted at a temperature of 25 °C (a temperature suitable for the seed germination of many species). However, our results confirmed that *Pinus bungeana* seed germination occurs at a comparatively cool temperature range of 15–20 °C. A final value of approximately 84%–92.67% was attained within this range, with the optimum temperature being 20 °C. The *Pinus bungeana* seeds were sensitive to warm temperatures, while when the temperature was raised to 25 °C and 30 °C, this caused the germination percentage to decrease sharply to 5% or less. This germination characteristic is representative of the eco-adaptability to their native habitats. In native habitats, *Pinus bungeana* seeds mature in October. Immediately then is the beginning of winter (in November, the night mean temperature is below freezing). In early spring, the mean temperature is 8–22 °C, snow begins to melt, resulting in the soil moisture content increasing, and then seeds begin to germinate. This behavior is similar to the germination requirements observed in some other coniferous trees. Skordilis and Thanos [33] found that the optimal germination temperature range of *Pinus halepensis* Mill. and *Pinus pinea* L. was 10–20 °C and that temperatures outside of that range retarded or inhibited germination. In comparatively low temperature range (15–20 °C), *Pinus brutia* Ten. [37] seeds show significantly higher germination than that at warmer temperatures, while the germination of *Widdringtonia whytei* Rendle [38] seeds exhibits a parabolic relation, with the optimal temperature being about 20 °C.

### 4.2. Germination Requirements after Storage Treatments

*Pinus bungeana* seed performance was evaluated for orthodox seeds, those resistant to a low water content, and those kept in cool storage for 1–3 years. The seeds retained high viability after dry cold storage for one year, and their germination of >80% at favorable temperatures is comparable to that of fresh seeds. Hereafter, seed viability (germination percentage) declined significantly with the increase of storage duration. In general, the germination percentage was decreased by about 10% for every additional year of storage, and the germination speed was reduced accordingly. Some freshly seeds commonly germinate within a particularly limited temperature range, which gradually widens with the increase of seed storage. Desert herb *Plantago coronopus* Linn. seeds germinated in a narrow cool temperature range (5–15 °C), while higher and quicker germination was obtained at a broad temperature scope (15–30 °C) when inflorescences were stored for one year in their native desert habitat [39]. Similarly, after storage, the demand for high temperature and light conditions to realize a greater germination percentage in fresh *Prosopis juliflora* (Swartz) DC. seeds was notably lowered [40]. Also, dry storage relieved seed dormancy and broadened the temperature demands for germination in *Bromus tectorum* Linn. [41], as well as light conditions in *Arabidopsis thaliana* (Linn.) Heynh. [42]. Nevertheless, storage did little to reduce the temperature request for the germination of *Pinus bungeana* seeds. Seeds stored for 1–3 years still achieved higher germination at lower temperatures (15 °C and 20 °C) yet obtained an extremely low germination percentage at warmer temperatures (25 °C and 30 °C). Dry cold storage did not effectively widen the temperature range of seed germination for *Pinus bungeana*.

### 4.3. The Reversibility of Germination Inhibition from High Temperatures

The literature shows that high temperatures inhibit seed germination, which is more obvious in the *Fraxinus* genus. In *Fraxinus ornus* subsp. *cilicica* (Lingelsh.) Yalt. [43] seeds, lower temperatures of 15 °C result in a higher germination percentage, and germinability declines rapidly at constant temperatures of 20 °C and 25 °C. *Fraxinus excelsior* Linn. [44] seeds have a higher germination rate at an alternating temperature of 5/15 °C, and germination generally decreases at a 20 °C or higher constant or alternating temperature. Incubation temperatures above the optimum always trigger secondary dormancy (thermo-dormancy). Piotto [45] pointed out this phenomenon, which had been found in *Fraxinus* spp. seeds at a constant temperature of 20 °C or above. In *Pinus bungeana* seeds, similarly, relatively higher temperatures 25 °C and 30 °C are beyond the favorable scope and also cause a sharp decrease in germination. However, the problem is elucidating whether

the ungerminated *Pinus bungeana* seeds enter into dormancy, like *Fraxinus*, or keep still at high temperatures. We found that all these ungerminated seeds germinate readily when transferred to lower temperatures, indicating that the germination inhibition of high temperatures on these pine seeds in particular is reversible. This temperature fluctuation is consistent with thermo-inhibition [8], and characteristically there is a loss or a sharp decrease in germination when the temperature is a little higher than the optimum temperature. The difference from the secondary dormancy is that those viable seeds germinate immediately without any pretreatment when the temperature is lowered. A similar performance of thermo-inhibition above 20 °C has been discovered in *Myrsine parvifolia* A. DC. as well [46]. Besides, thermo-inhibition phenomena have also been observed in two herbs, namely, *Lactuca sativa* Linn. and *Hordeum distichon* Linn. [47]. In these cases, the seeds are non-dormant but keep still without germinating to deal with the unfavorable external environment (high temperature).

### 4.4. Seed Tissues Involved in Inhibited Germination at High Temperatures

According to the present findings, when the seed coat was taken from the seed, the amount of oxygen reaching the embryo was greatly increased, and any inhibitory substance that might be present in the seed coat that could hinder germination would be excluded, but most of the *Pinus bungeana* seeds still did not germinate, suggesting that the seed coat is not primarily responsible for the failure of seed germination at high temperatures (25 °C). Before research on the water absorption of a pine species, namely, *Pinus sylvestris* L. [48], the nucellar membrane had rarely been noticed, for which the seed tissue of this coniferous tree is usually flimsy and frangible and does not seem to hinder germination. In our study, only 34.67% of seeds germinated after removing the seed coat, while the germination percentage reached 84% when the nucellar membrane was subsequently taken off. These results suggest that the nucellar membrane contributes significantly to inhibiting *Pinus bungeana* seed germination at high temperatures. Furthermore, there might be a synergy between the nucellar membrane and seed coat in this inhibition. However, the seeds of *Pinus bungeana* had no germination barrier under the lower temperature conditions of 15 °C and 20 °C.

### 4.5. The Role of ABA Played in Germination at High Temperatures

The enclosing tissues of the embryos generally inhibit seed germination by interfering with water absorption and gas exchange, mechanically restricting radicle protrusion and impeding the escape of inhibitors or containing inhibitory substances themselves. In *Pseudotsuga menziesii* (Mirb.) Franco seeds, the seed coat interferes with gas exchange and partially blocks the extravasation of the inhibitors to affect germination [49], while the nucellar membrane is an effective barrier for the water absorption of *Pinus elliottii* Engelm. seeds [50].

As mentioned above, the temperature conditions were very important for the germination of *Pinus bungeana* seeds. At high temperatures, the seed tissue nucellar membrane played a dominant role in impeding seed germination. Microstructural observation showed that the nucellar membrane was longitudinally ruptured in the late incubation period at 20 °C (after 8 days). Other than this, there was no significant difference in the inner and outer surfaces and cross-section microstructure of the nucellar membranes at 20 °C and 25 °C. This could be explained by the fact that after one week at 20 °C, the seeds began to germinate and their water content increased rapidly, then the megagametophyte continued to swell, causing the nucellar membrane that wraps it to rupture. However, at 25 °C, the temperature conditions were too high for *Pinus bungeana* seed germination, and those imbibed seeds were in a 'stationary' state when the temperature was slightly above what was supraoptimal for their germination. Then, the water content of the megagametophyte was no longer elevated and the nucellar membrane remained intact. Thus, this 'inhibition' mechanism of seed germination in *Pinus bungeana* at high temperatures should not be attributed to the structure of nucellar membrane imposed, and the inhibitors in it might be responsible for this.

There are many endogenous chemicals (mostly hormonal) that inhibit germination in seeds, while ABA is most often relevant to the application and maintenance of high-temperature inhibition

germination (i.e., thermo-inhibition) [51,52]. The literature on the roles of ABA in thermo-inhibition in conifer seeds is finite, but ABA being involved in controlling seeds dormancy and the germination of the conifer species has been documented. In the course of the dormancy release of *Chamaecyparis nootkatensis* (D. Don) Spach seeds, the embryos exhibited a change (a two-fold reduction) in ABA turnover [53]. Similarly, cold treatment breaking *Pseudotsuga menziesii* (Mirb.) Franco seed dormancy was associated with variation of the ABA content and sensitivity, and full germination was obtained in the presence of 1 µM of fluridone (a substance that inhibits the synthesis of ABA) [49,54].

Generally, the ABA content in seeds that germinate at low temperatures is reduced more rapidly than that at high temperatures [20]. In the *Pinus bungeana* seeds, under the favorable temperatures of 20 °C, the catabolism of ABA exceeded its biosynthesis and the ABA content in the nucellar membrane decreased significantly. When it dropped to a certain threshold (15 ng/g (FW)), germination occurred. In comparison, at 25 °C, germination inhibition was related to the high content of ABA, where the high temperatures facilitate ABA biosynthesis, yet its inactivation is prevented via metabolism and conjugation, where ABA levels in the nucellar membrane then increase, causing the failure of radicle protrusion and probably hypocotyl elongation. This effect of ABA on germination in thermo-inhibition has been confirmed in many other plants, including *Tagetes minuta* L. [55], *Solanum lycopersicum* Lam. [8] and *Arabidopsis thaliana* (Linn.) Heynh. [56], etc. However, the rate of inhibitory substances or ABA catabolism depends on the oxygen concentration, and the oxygen availability in an imbibed seed is largely determined by low temperatures, which improve the dissolved oxygen levels in nearby water [57]. This could clarify how the ABA levels in the nucellar membrane decreased more quickly at 20 °C than at the unfavorable temperature of 25 °C. Simultaneously, the presence of the nucellar membrane might lead to the embryo existing in a hypoxic microenvironment, which, conversely, facilitates ABA synthesis and inhibits its inactivation. That is, the presence of the nucellar membrane could interfere with the accumulation of ABA in *Pinus bungeana* seeds at high temperatures. Also, seeds usually are much more sensitive to ABA at a high temperature, at which germination is strongly inhibited, rather than at a temperature at which seeds germinate easily [58]. This might also play a role in preventing seeds from germinating at high temperatures of 25 °C.

## 5. Conclusions

The light condition was not a limiting factor for the radicle protrusion of *Pinus bungeana* seeds, but the temperature was the critical factor for germination. Within the tested temperature range, the optimum temperature for germination was 20 °C, and 15 °C was only slightly less favorable. Under suitable temperatures, a high germination percentage (approximately 90%) was obtained. However, higher incubation temperatures led to a drastic decline in germination and triggered thermo-inhibition. The germination of this species is limited to a straight temperature scope, which cannot be effectively widened via dry cold storage. Moreover, the embryo coverings layers, especially the nucellar membrane, contributed most in inhibiting seed germination at high temperatures, and the regulation of ABA content in the nucellar membrane is key for the thermo-inhibition of *Pinus bungeana* seeds. Nevertheless, it should be noted that ABA might also exist in other seed tissues (mainly seed coats), which could inhibit germination to a certain extent. In addition to ABA, there might be some other chemicals in the nucellar membrane that contribute to germination inhibition at high temperatures. Further research will be carried out to analyze these issues, which will help us to gain a better comprehension of the thermo-inhibition mechanism of *Pinus bungeana* seeds.

**Author Contributions:** C.G., Y.S., and F.S. partook in the discussion and designed the experiment; Y.S. prepared the experiment materials; C.G. conducted the laboratory analyses and wrote the paper. All authors have read and agreed to the published version of the manuscript.

**Funding:** This research was funded by the Priority Academic Program Development of Jiangsu Higher Education Institutions (PAPD).

**Conflicts of Interest:** The authors declare no conflict of interest.

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
