# Peer review of "Effect of Temperature, Light, and Storage Time on the Seed Germination of Pinus bungeana Zucc. ex Endl.: The Role of Seed-Covering Layers and Abscisic Acid Changes"

_forests, doi:10.3390/f11030300_

Round 1
Reviewer 1 Report
The manuscript of Congcong Guo and co-workers present an interesting work on conditions suitable for germination of Pinus bungeana seeds. In my opinion, obtained results are important for understanding the mechanisms underlying of seed dormancy in coniferous plants. The data are well documented and presented. I have only a few doubts and comments for Authors consideration:
- Lines 189-191. Are the given SG and MGT values correct? The values do not comply with the data shown in Table 2, or maybe I am wrong?
- Line 229. Please explain what TTC is. There is also no information about this test in the Materials and Methods section.
- Figure 4. The black asterisks and scale bars are not well visible in the micrographs.
Author Response
- Comment: Lines 189-191. Are the given SG and MGT values correct? The values do not comply with the data shown in Table 2, or maybe I am wrong?
Response: I'm sorry, this is a mistake in our writing. The given SG and MGT values (lines 189-191) were written incorrectly because our carelessness, now they have been modified to the correct values which complied with the data shown in Table 2.
- Comment: Line 229. Please explain what TTC is. There is also no information about this test in the Materials and Methods section.
Response: In section 2.1 Seed Materials, we have added the explanation of TTC. It was shown in the text as follows: “…As determined by the 2,3,5‐triphenyl tetrazolium chloride (TTC) method [29], the viability of fresh seeds was…”
- Comment: Figure 4. The black asterisks and scale bars are not well visible in the micrographs.
Response: We have adjusted the size of the asterisks and scales bars in the micrographs of figure 4, both of them are easier to read now.

Reviewer 2 Report
Title: Effect of Temperature, Light and Storage on Seed Germination of Pinus bungeana and Role of Covering Layers and Changes in Concentration of Abscisic Acid brings important information but is too long, so it can be improved as “Effect of Temperature, Light and Storage time on Seed Germination of Pinus bungeana: Role of Seed-Covering Layers and abscisic acid (ABA) changes”; Abstract is concise but language needs to be revised by a native speaker; about Introduction the background and problem are clearly stated; some more relevant papers can be added; Materials and Methods and Results are generally acceptable; Conclusions are in line by the obtained results; References are appropriate but they could be updated; Tables and Figures can be improved, especially Tab. 3 where in the column ‘Rot Rate’ it is not clear what it refers to; Figures 2 and 6 are not clear. All the text require moderate English changes to improve the paper.
I have made a few suggestions in "Minor Considerations". The following points may be considered while revising the manuscript:
- in lines 44, the authors give general information about the ability of a viable seeds to germinate when external environmental conditions are favourable, so many species require an appropriate temperature range. Surely, temperature and light are important ecological factors for seed germination. I suggest to add this following specific work to improve the paper:
Cristaudo, A.; Gresta, F.; Restuccia, A.; Catara, S.; Onofri, A. 2016. Germinative response of redroot pigweed (Amaranthus retroflexus L.) to environmental conditions: Is there a seasonal pattern? Plant Biosystems, Volume 150, Issue 3, Pages 583-591, where is reported that seed germination behaviour in common weed is not independent of the time of the year in which seeds are produced and is due to both the environmental conditions experienced by the mother plant during seed maturation and those experienced by seeds after seed set.
- in lines 47-50 to improve the references, I suggest to add this paper:
Gresta, F.; Cristaudo, A.; Onofri, A.; Restuccia, A.; Avola, G. 2010. Germination response of four pasture species to temperature, light, and post-harvest period. Plant Biosystems, Volume 144, Issue 4, Pages 849-856, where the effect of temperature, light, and post-harvest period, and their interactions, on seed germination ecology of four common pasture species in the Mediterranean environment was studied; these ecological requirement may be regarded as an adaptation strategy to ensure optimal conditions for seedling development and survival in Mediterranean species.
- In line 109 the authors report that the viability of the fresh seeds is >90%; did they carried out germination or viability tests? Please re-word for clarity.
- In line 117 are not reported the model of incubator, please add the information
- In line 119-120 it has not been explained how the controls in dark conditions are carried out, please explains how the controls take place
- in line 215 and in line 273 the Figures 2 and 6 must be improved
- in line 242 the table 3 reports as column ‘Rot rate’, please explain what does it mean. It is not reported in the main text.
- in line 366 the authors give a series of questions in the Section “Discussion” of results. Delete these questions because you should already give some answers
Author Response
- Comment: Title: Effect of Temperature, Light and Storage on Seed Germination of Pinus bungeana and Role of Covering Layers and Changes in Concentration of Abscisic Acid brings important information but is too long, so it can be improved as “Effect of Temperature, Light and Storage time on Seed Germination of Pinus bungeana: Role of Seed-Covering Layers and abscisic acid (ABA) changes”;
Response: This is a valuable suggestion, and the title has been improved as “Effect of Temperature, Light, and Storage Time on the Seed Germination of Pinus bungeana Zucc. ex Endl.: The Role of Seed-Covering Layers and Abscisic Acid Changes”
- Comment: All the text require moderate English changes to improve the paper.
Response: We have sent the manuscript to MDPI English Editing Service for language service before submitting the revised version. These revisions are marked in red in the paper.
- Comment: in lines 44, the authors give general information about the ability of a viable seeds to germinate when external environmental conditions are favourable, so many species require an appropriate temperature range. Surely, temperature and light are important ecological factors for seed germination. I suggest to add this following specific work to improve the paper:
Cristaudo, A.; Gresta, F.; Restuccia, A.; Catara, S.; Onofri, A. 2016. Germinative response of redroot pigweed (Amaranthus retroflexus L.) to environmental conditions: Is there a seasonal pattern? Plant Biosystems, Volume 150, Issue 3, Pages 583-591, where is reported that seed germination behaviour in common weed is not independent of the time of the year in which seeds are produced and is due to both the environmental conditions experienced by the mother plant during seed maturation and those experienced by seeds after seed set.
Response: We have added the reference in our paper and updated the references simultaneously. It was shown in the text as follows: “…And most species require an appropriate temperature range or alternate temperature mode to achieve maximum germination. In weeds, the germination behavior of seeds is also related to the time of seed produced and the moment elapsed from the seed settling. This behavior is owed to the environmental conditions undergone by the mother plant during seed maturation and those undergone by the seeds after settling [3]. The germination percentage usually increases linearly with temperature up to an optimal temperature...”
- Comment: in lines 47-50 to improve the references, I suggest to add this paper:
Gresta, F.; Cristaudo, A.; Onofri, A.; Restuccia, A.; Avola, G. 2010. Germination response of four pasture species to temperature, light, and post-harvest period. Plant Biosystems, Volume 144, Issue 4, Pages 849-856, where the effect of temperature, light, and post-harvest period, and their interactions, on seed germination ecology of four common pasture species in the Mediterranean environment was studied; these ecological requirement may be regarded as an adaptation strategy to ensure optimal conditions for seedling development and survival in Mediterranean species.
Response: We have added the reference in the corresponding line and updated the references simultaneously. It was shown as follows: “…To prevent seedlings from being damaged after germination, physiological reactions may occur in seeds to cope with the high-temperature environment in which they are placed. This ecological requirement can be considered as an adaptation strategy to guarantee favorable conditions for seedling development and survival in some species [7]. Then, these seeds will cease germination…”
- Comment: In line 109 the authors report that the viability of the fresh seeds is >90%; did they carried out germination or viability tests? Please re-word for clarity.
Response: The seed viability was determined by the TTC test, this sentence has been rewritten as: “…As determined by the 2,3,5‐triphenyl tetrazolium chloride (TTC) method [29], the viability of the fresh seeds collected in different years was all >90%...”
- Comment: In line 117 are not reported the model of incubator, please add the information
Response: We have added the detail information of the incubator, the sentence was expressed as follows: “…Then the boxes were distributed in a GTOP-300D incubator (Zhejiang Top Cloud-agri Technology Co., Ltd., China) at random with a constant temperature…”
- Comment: In line 119-120 it has not been explained how the controls in dark conditions are carried out, please explains how the controls take place
Response: Now in our paper, we have explained how the controls in dark conditions are carried out, it was shown as: “…the dark treatment condition was achieved by covering the germination boxes with tinfoil. Under dark conditions, the germination was assessed in a room equipped with a green light. Germinated seeds were counted…”
- Comment: in line 215 and in line 273 the Figures 2 and 6 must be improved
Response: To make Figure 2 more clear and readable, we have changed its form to a histogram, and added the unit after the title on the ordinate; Figure 6 shows the relationship between abscisic acid and germination percentage in the form of a scatterplot. But the dots on the left of the figure are a bit too dense and don't seem to be so delicate. We also made other attempts before, such as using unequal distance axes or dual axes or using a combination of histograms and line charts, but these results are not ideal. That is because that the germination percentage is almost zero under the conditions of high abscisic acid concentration (25 ° C, early days of 20 ° C), these dots gather together on the figure and look dense, but this situation does not seem to be improved by changing the chart format. So after careful consideration, we think that the current scatterplot might be a relatively appropriate form for this part.
- Comment: in line 242 the table 3 reports as column ‘Rot rate’, please explain what does it mean. It is not reported in the main text.
Response: ‘Rot rate’ was mentioned in the last sentence of section 2.5, but we did not explain what it means. Now we have added its explanation after the ‘Rot rate’, the sentence was rewritten as: “…The germinated seeds were scored daily for 30 days, and the TGP and rot rate (percentage of rotten seeds) was determined at the end of each experiment…” Besides, we improved the unit problem in tables 2 and 3 and added the SD in table 3.
- Comment: in line 366 the authors give a series of questions in the Section “Discussion” of results. Delete these questions because you should already give some answers
Response: This is a valuable comment, and we have deleted these questions in the last part of section 4.4

This manuscript is a resubmission of an earlier submission. The following is a list of the peer review reports and author responses from that submission.